# A Generalised Multifrequency PWM Strategy for Dual Three-Phase Voltage Source Converters

**Jose A. Riveros** [1,2,*] **, Joel Prieto** [3] **, Marco Rivera** [1] **, Sergio Toledo** [1,4] **and Raúl Gregor** [4]

1   Facultad de Ingeniería, Universidad de Talca, Curicó 3341717, Chile; marcoriv@utalca.cl (M.R.); stoledo@utalca.cl (S.T.)
2   Facultad Politécnica, Universidad Nacional de Asunción, San Lorenzo 2111, Paraguay
3   Facultad de Ciencias de la Ingeniería, Universidad Paraguayo Alemana, San Lorenzo 2160, Paraguay; joel.prieto@upa.edu.py
4   Facultad de Ingeniería, Universidad Nacional de Asunción, Luque 2060, Paraguay; rgregor@ing.una.py
*   Correspondence: jriveros@utalca.cl; Tel.: +56-752-315-470

**Abstract:** Pulse width modulation (PWM) strategies for the control of asymmetrical six-phase drives have been widely studied since the beginning of this century. Nevertheless, space vector modulation (SVM) techniques with multifrequency voltage injection for the control of all the degrees of freedom of the multiphase model is still a subject under research. This paper deals with this topic and introduces a generalised PWM method for a two-level voltage source converters. The architecture was derived by extending a three-phase modulator proposed as an alternative to the widely studied SVM. The proposal computes the duty times straightforwardly with a fast algorithm based on an analytical solution of the voltage-time modulation law. Theoretical derivations supported by experimental results demonstrate the proper synthesis of the multifrequency target voltage in the linear modulation region as well as good frequency behaviour of the presented modulation strategy.

**Keywords:** multiphase drives; pulse width modulation; dc-ac power converters

## 1. Introduction

Multiphase technology has become one of the most attractive subjects within the electric drives research area [1]. Since the beginning of this century, numerous publications in journals and dedicated sessions of conferences have reported innovative exploiting of their additional number of degrees of freedom respect to the conventional topology [2]. Thus, high-performance control strategies have been developed to take advantage of the most promoted features such as fault-tolerant, efficient electromechanical energy conversion and distribution of the current stress into more than three phases [3–5]. These are commonly designed to be used in applications such as electric vehicles, ship propulsion, renewable energy generation and high-power industry [6]. The development of modulation and control schemes capable of regulating the entire multiphase modelling, compounded by multiple two-dimensional subspaces [7], is still one of the main topics [3,4]. Implementations with low computational cost are the desired and most impacting result in this task, considering that the hardware barriers (digital controllers, topologies and power switches) have been overcome with the last advancements and maturity achieved in other involved fields [2–4].

The asymmetrical dual three-phase machines are one of the most considered designs within multiphase systems [1]. This proposal is compounded by two sets of three-phase windings electrically shifted by 30 degrees in order to attain the best torque ripple behaviour. This topology presents compatibility with the three-phase power converters available in the market, which can be associated to become a six-phase voltage source converter (VSC) and supply the described multiphase machine.

Moreover, the sets of windings can be connected with single or double neutral-point formats. The first configuration provides enhancements in the electromagnetic torque generation [8] and better post-fault characteristics [9]. On the other hand, the isolated neutrals arrangement enables the best dc-bus utilisation [10] and less susceptibility to the low-order stator current harmonics (the triplen components are annulled) [11]. This proposal currently meets the requirements to replace the three-phase drives in high-power applications. Nevertheless, new advancements in the control and modulation strategies will contribute to the gradual adoption of the multiphase technology in non-conventional uses.

The pulse width modulation (PWM) methods for the asymmetrical dual three-phase drives have been widely covered in different researches [12–16]. These techniques are developed by using the vector space decomposition (VSD) approach [12]. Thus, the multiphase inverter modelling is represented in multiple complex planes. The main subspace held the fundamental frequency component, whereas the harmonics and zero-sequence variables are located in the complementary planes. The resulting space can be organised in twelve [13] or twenty-four [14] sectors to develop the voltage space vector modulation (SVM) strategy. A higher number of sectors enhances the performance in terms of flux harmonic distortion factor at the expense of a higher computational burden [17]. For the overmodulation region, the implementation of [15] employs two three-phase SVM with a competitive classification algorithm, and a minimum harmonic distortion modulator was presented in [16]. All the reviewed proposals had been developed considering a single frequency reference voltage (zero command voltage in the harmonics' subspace).

The surveyed schemes do not provide the highest performance in real multiphase drives, because they lead unwanted low-order stator current harmonics [18]. These are caused by the small constructive asymmetries of the electric machine and, in higher proportion, by the non-linear effects of the switching dead-time. Injection of current in the complementary subspaces (possible with references harmonics voltage different than zero) of the multiphase modelling is required to overcome this drawback. For this reason, multifrequency modulation schemes are an interesting research topic in this field. However, this subject has been barely covered in the literature considering the six-phase VSCs and implementations report the use of double zero-sequence injection with the carrier-based approach to accomplish with this goal [2]. Recently, two proposals were assessed in [19,20] as an extension of the five-phase modulator presented in [21]. Two voltage-time equation systems (one for the fundamental frequency and another for the harmonics' plane) are solved, and a time-multiplexing has been used for the application of these solutions within a sampling period. The technique could employ up to eight active voltage space vectors (twice the necessary number to control both planes of the model) to synthesise independent voltage outputs at two different frequencies. Additionally, an asymmetric switching pattern (characterised by higher current ripples and more complex implementation) was reported in the validations of this modulation architectures with simulation results.

A new PWM method is introduced in this work. The strategy controls all the degrees of freedom of the six-phase VSC by employing a generalised analytical solution of the modulation law. The technique is an extension of the three-phase modulator recently developed in [22]. This approach has been extrapolated by adding the harmonics' subspace with the VSD theory and arranging the model to attain two decoupled three-phase modulators. These last are commanded by two auxiliary voltages defined straightforwardly from the original reference voltage vectors. The configuration allows the operation in the multifrequency mode with a low computational cost algorithm. The inputs of the architecture are the reference components in the stationary reference frame and the zero-sequence control signal to compute the duty cycles of the inverter's legs. Consequently, the magnitude and position (sector) are not necessary, avoiding the high computational effort required for operators such as squared root or trigonometric functions. Continuous switching and healthy operation of the inverter within the linear modulation interval is the scope for the introduction of the proposal.

This paper is organised as follows. The next section reviews the modelling of the system with emphasis in the output voltages. Then, the developed modulation strategy is detailed in Section 3.

Next, Section 4 discusses the experimental validation for the time- and frequency-domain. The last part presents the conclusions obtained after the evaluation process.

## 2. Modelling of the Dual Three-Phase Voltage Source Converter

The two-level six-phase VSC is a popular topology within the research area because of their promising features [4]. An electrical diagram of this design is shown in Figure 1. The inverter is power-supplied through the dc-bus with a $V_{dc}$ input voltage. This is processed by means of an arrangement of six legs, which in turns are composed by two power semiconductors in series. These last must operate in the complementary conduction mode to avoid damaging currents. Thus, the switching state $S_j$ ($j = \{a, b, \ldots, f\}$) can be modelled with a bit signal, where $S_j = 1(0)$ indicates that the top(bottom) power switch of the leg $j$ is activated. The variables of the three-phase sets are designated with the subscripts *a-b-c* and *d-e-f*, respectively. The switching functions provide different phase voltages $V_j$, and a total of 64 combinations can be generated and calculated as follows [11]:

$$\begin{bmatrix} V_a \\ V_d \\ V_b \\ V_e \\ V_c \\ V_f \end{bmatrix} = \frac{V_{dc}}{3} \begin{bmatrix} 2 & 0 & -1 & 0 & -1 & 0 \\ 0 & 2 & 0 & -1 & 0 & -1 \\ -1 & 0 & 2 & 0 & -1 & 0 \\ 0 & -1 & 0 & 2 & 0 & -1 \\ -1 & 0 & -1 & 0 & 2 & 0 \\ 0 & -1 & 0 & -1 & 0 & 2 \end{bmatrix} \begin{bmatrix} S_a \\ S_d \\ S_b \\ S_e \\ S_c \\ S_f \end{bmatrix} \tag{1}$$

The scalar model of the VSC derived in (1) can be simplified to facilitate the development of the modulation strategy by applying the VSD approach [12]. The linear transformation with invariant magnitude format is detailed below:

$$\begin{bmatrix} V_\alpha \\ V_\beta \\ V_x \\ V_y \\ V_{0^+} \\ V_{0^-} \end{bmatrix} = \frac{1}{3} \begin{bmatrix} 1 & C_\delta & -S_\delta & -C_\delta & -S_\delta & 0 \\ 0 & S_\delta & C_\delta & S_\delta & -C_\delta & -1 \\ 1 & -C_\delta & -S_\delta & C_\delta & -S_\delta & 0 \\ 0 & S_\delta & -C_\delta & S_\delta & C_\delta & -1 \\ 1 & 0 & 1 & 0 & 1 & 0 \\ 0 & 1 & 0 & 1 & 0 & 1 \end{bmatrix} \begin{bmatrix} V_a \\ V_d \\ V_b \\ V_e \\ V_c \\ V_f \end{bmatrix} \tag{2}$$

with $C_\delta$ and $S_\delta$ being the cosine and sine operators of $\delta = \pi/6$, respectively. These are the components in an orthogonal and stationary reference frame. Furthermore, the fundamental frequency along with the $12k \pm 1$ ($k = 1, 2, \ldots$) order harmonics are held in the $\alpha$-$\beta$ plane, the $6k \pm 1$ order harmonics are mapped into the $x$-$y$ subspace and the zero-sequence components are projected in the remaining space. The decoupled components can be also calculated with the switching functions by combining (1)–(2). The result with the normalised voltages respect to $V_b = V_{dc}/2$ is the following:

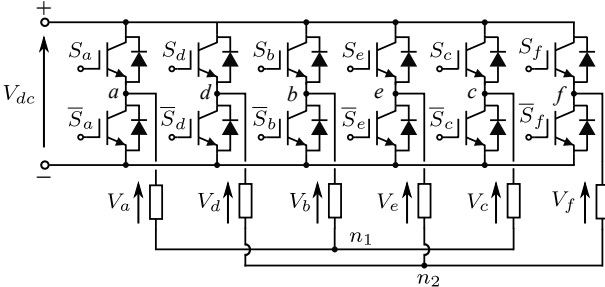

**Figure 1.** Six-phase voltage source converter (VSC) power-supplying a dual three-phase load with isolated neutrals.

$$
\begin{bmatrix} v_\alpha \\ v_\beta \\ v_x \\ v_y \\ v_{0^+} \\ v_{0^-} \end{bmatrix} = \frac{2}{3} \begin{bmatrix} 1 & C_\delta & -S_\delta & -C_\delta & -S_\delta & 0 \\ 0 & S_\delta & C_\delta & S_\delta & -C_\delta & -1 \\ 1 & -C_\delta & -S_\delta & C_\delta & -S_\delta & 0 \\ 0 & S_\delta & -C_\delta & S_\delta & C_\delta & -1 \\ 0 & 0 & 0 & 0 & 0 & 0 \\ 0 & 0 & 0 & 0 & 0 & 0 \end{bmatrix} \begin{bmatrix} S_a \\ S_d \\ S_b \\ S_e \\ S_c \\ S_f \end{bmatrix} \tag{3}
$$

where $v_m = V_m/V_b$ are the normalised voltage components of the $m$-axis ($m = \{\alpha, \beta, x, y, 0^+, 0^-\}$). This approach cannot be used to study the zero-sequence components, but these are a degree of freedom of the system that could be added later to improve some performance characteristic [22]. For this reason, the $0^+$ and $0^-$ component are omitted from the derivation of the modulation scheme henceforth. The modelling of the VSC is also described by current equations and the common-mode voltage. These are not included for the sake of simplicity in the introduction of the proposed technique.

## 3. Multifrequency Pulse Width Modulation Algorithm

A modulation space for the dual three-phase VSC described by the switching signals can be derived as an extension of the approach introduced in [22]. Hence, expanding the matrix operators of the modelling (3) derived with the VSD theory, the following result is achieved:

$$
\begin{aligned}
v_\alpha &= (2/3)(S_a + C_\delta S_d - S_\delta S_b - C_\delta S_e - S_\delta S_c) \\
v_\beta &= (2/3)(S_\delta S_d + C_\delta S_b + S_\delta S_e - C_\delta S_c - S_f) \\
v_x &= (2/3)(S_a - C_\delta S_d - S_\delta S_b + C_\delta S_e - S_\delta S_c) \\
v_y &= (2/3)(S_\delta S_d - C_\delta S_b + S_\delta S_e + C_\delta S_c - S_f)
\end{aligned} \tag{4}
$$

Then, by following the procedure of [22], the modulation law is achieved by integrating (4) over time within a sampling period $T_s$. The resulting voltage-time system with the normalised duty cycles $t_j$ (period of time in which $S_j$ is set to 1 within a sampling period) respect to $T_s$ and the reference voltages $v_m^*$ is the following:

$$
\begin{aligned}
v_\alpha^* &= (2/3)(t_a + C_\delta t_d - S_\delta t_b - C_\delta t_e - S_\delta t_c) \\
v_\beta^* &= (2/3)(S_\delta t_d + C_\delta t_b + S_\delta t_e - C_\delta t_c - t_f) \\
v_x^* &= (2/3)(t_a - C_\delta t_d - S_\delta t_b + C_\delta t_e - S_\delta t_c) \\
v_y^* &= (2/3)(S_\delta t_d - C_\delta t_b + S_\delta t_e + C_\delta t_c - t_f)
\end{aligned} \tag{5}
$$

The modulation law can be arranged by means of elemental operations in order to achieve an equivalent linear equation system (the solution is not affected). Two decoupled and simpler modulation laws are obtained by combining appropriately the equations of (5). The simplified result is summarised as follows:

$$
\begin{aligned}
(v_\alpha^* + v_x^*) &= (4/3)(t_a - S_\delta t_b - S_\delta t_c) \\
(v_\beta^* - v_y^*) &= (4/3)(C_\delta t_b - C_\delta t_c)
\end{aligned} \tag{6}
$$

$$
\begin{aligned}
-(v_\beta^* + v_y^*) &= (4/3)(t_f - S_\delta t_d - S_\delta t_e) \\
(v_\alpha^* - v_x^*) &= (4/3)(C_\delta t_d - C_\delta t_e)
\end{aligned} \tag{7}
$$

The systems of (6) and (7) are two independent three-phase modulation spaces according to the approach of [22]. Consequently, they can be implemented with the fast algorithm developed in the cited work. A description of the method is included in Appendix A. In this scheme, the left-side of the identities are the stationary reference voltages. Then, the modulators should be commanded with the following auxiliary target signals:

$$v_{d1}^* = v_\alpha^* + v_x^*$$
$$v_{q1}^* = v_\beta^* - v_y^*$$
$$v_{d2}^* = -(v_\beta^* + v_y^*)$$
$$v_{q2}^* = v_\alpha^* - v_x^*$$

(8)

Additionally, notice that the model of the second modulator, see the first equation of (7), indicates that the first output of this block controls the $S_f$ switching signal, whereas the remaining ought to be linked to $S_d$ and $S_e$, respectively.

The block diagram of the proposed strategy is depicted in Figure 2. The two three-phase modulators are commanded by the references voltages defined in (8) along with the zero-sequence control signals $\lambda_1$ and $\lambda_2$. These last are set to 0.50 to provide the SVM operation, see Table A1, as an introduction of the method. Thus, the first modulator, MOD1, is configured with the references $v_{d1}^*$ and $v_{q1}^*$ to compute the duty cycles of the legs controlled by the $S_a$, $S_b$ and $S_c$ switching signals of the six-phase VSC. The MOD2 is commanded with $v_{d2}^*$ and $v_{q2}^*$, while the transposition of the switching signal previously described is implemented. The three-phase techniques attain the duty times with Algorithm A1, detailed in the appendix. This information is employed by a six-channel digital PWM peripheral, which works with up/down counters and comparators to generate the gating signals. Theses activate/deactivate the power switches to synthesise the multifrequency output voltage.

Let us consider the following example to illustrate the operation of the proposed technique. The references voltages are $v_\alpha^* = 0.3653$, $v_\beta^* = 0.9309$, $v_x^* = 0.0956$, and $v_y^* = -0.0295$. Hence, the auxiliary references voltages defined by (8) are:

$$v_{d1}^* = v_\alpha^* + v_x^* = 0.3653 + 0.0956 = 0.4609$$
$$v_{q1}^* = v_\beta^* - v_y^* = 0.9309 + 0.0295 = 0.9604$$
$$v_{d2}^* = -(v_\beta^* + v_y^*) = -(0.9309 - 0.0295) = -0.9014$$
$$v_{q2}^* = v_\alpha^* - v_x^* = 0.3653 - 0.0956 = 0.2697$$

(9)

Then, the operation of the three-phase modulators with $\lambda_1 = \lambda_2 = 0.50$ is summarised as follows:

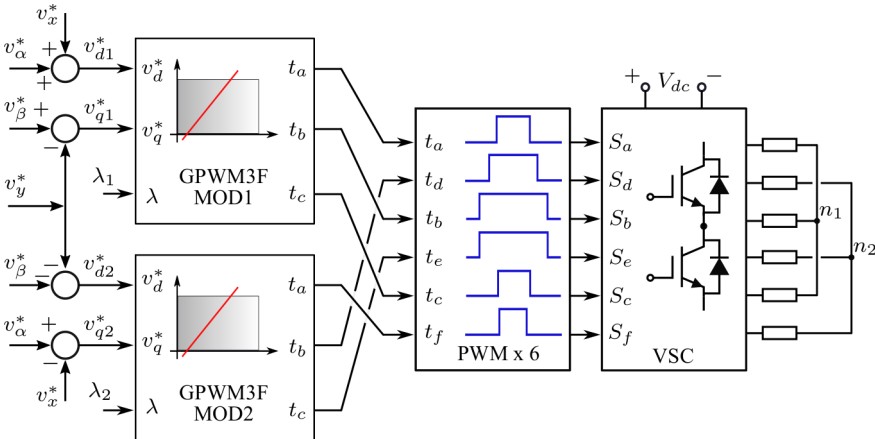

**Figure 2.** Generalised pulse width modulation (PWM) for asymmetrical six-phase VSCs.

MOD1:

$\tau_d = C_\delta \cdot |v_{q1}| = 0.8317$

$u^* = C_\delta^2 \cdot v_{d1} + S_\delta \cdot \tau_d = 0.7615$

$0 < u^* \leq \tau_d$, then:

$\tau_{11} = u^* = 0.7615$

$a = 1 - \tau_d = 0.1683$

$t_a = \tau_{11} + a \cdot \lambda_1 = 0.8457$

$t_b = t_a - C_\delta \cdot (C_\delta \cdot v_{d1}^* - S_\delta \cdot v_{q1}^*) = 0.9159$

$t_c = t_b - C_\delta \cdot v_{q1}^* = 0.0841$

MOD2:

$\tau_d = C_\delta \cdot |v_{q2}| = 0.2336$

$u^* = C_\delta^2 \cdot v_{d2} + S_\delta \cdot \tau_d = -0.5593$

$-C_\delta \leq u^* \leq 0$, then:

$\tau_{11} = 0$

$a = 1 + u^* - \tau_d = 0.2072$

$t_f = \tau_{11} + a \cdot \lambda_2 = 0.1036$

$t_d = t_f - C_\delta \cdot (C_\delta \cdot v_{d2}^* - S_\delta \cdot v_{q2}^*) = 0.8964$

$t_e = t_d - C_\delta \cdot v_{q2}^* = 0.6628$

Notice that the operation of the proposal is simple and controls all the degree of freedom of the multiphase VSC. The output (duty cycles) is suitable for the implementation of the algorithm with PWM peripheral of the digital controllers. All these features are promising for the promotion of the multiphase technology in the industrial sector.

## 4. Experimental Validation and Discussions

The objective of this section is to provide the experimental proofs of the proper operation of the developed method. The architecture is composed by two three-phase generalised PWM blocks, whose performance was assessed only with single frequency tests. In this work, this scheme is commanded by reference voltages composed of a mix between two independent magnitudes and frequencies. Then, the proper generation of the output voltage in the time and frequency domains ought to be verified to demonstrated its viability for multiphase applications.

The developed modulator is assessed in the experimental test rig indicated in Figure 3. This is a six-phase VSC built with the FGH80N60FDTU IGBT. The load employed for the tests is a resistor-inductor, which is the most common behaviour in multiphase applications. The parameters of the experimental setup are detailed in Table 1. The dc-bus voltage is attained with a three-phase diode bridge rectifier power-supplied by a variac to control the output voltage, which is filtered by two capacitor in series located in the dc-side. The voltage $V_{dc}$ is regulated to 100 (V) in order to avoid overcurrents during the tests with the maximum dc-bus utilisation. The frequency index ($m_f = f_s / f_1$) defines the sampling frequency $f_s = 1.5$ (kHz) and period $T_s = 1/f_s$, while the fundamental frequency is 50 (Hz). The controller is the experimenter kit of Texas Instruments based on the digital signal processor (DSP) TMS320F28335, while the commutations and protections are in charge of the Nexys 3 Spartan-6 FPGA Trainer Board. The algorithm is programmed in the DSP that send the six PWM signals to the FPGA. This last generates the twelve switching states with a dead-time of 2 ($\mu$s) and stops the operation under the presence of a trip-zone alarm. An optical link between the digital and power stages is implemented to reduce the electromagnetic interference impact. Two three-phase VSCs interconnected by the dc-bus in a modular design supply the load. The architecture of the prototype is flexible for future expansion to study drives with a higher number of phases or implement modulation techniques with unconventional switching strategies. The Keysight DSOX3024T oscilloscope is employed to acquire the measurements along with the Fluke i3000s Flex-24 ac current clamp and the Elditest differential voltage probe GE 8100.

**Table 1.** Parameters for the experimental tests.

| Parameter | Unit | Value |
|---|---|---|
| Resistance, $R$ | ($\Omega$) | 10 |
| Inductance, $L$ | (H) | 10 m |
| Dc-bus Voltage, $V_{dc}$ | (V) | 100 |
| Fundamental frequency, $f_1$ | (Hz) | 50 |
| Frequency index, $m_f$ | | 30 |

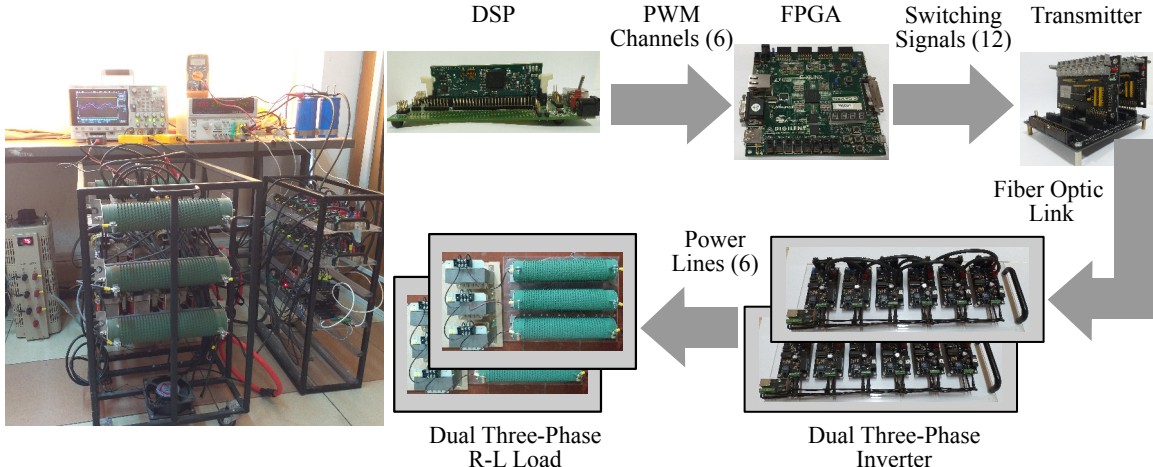

**Figure 3.** Experimental setup for the validation process.

The magnitude of the reference voltages are controlled with the fundamental and harmonic modulation indexes; $m_1$ and $m_2$, respectively. The second voltage is set with the fifth-order harmonic frequency (250 Hz) in order to inject this component in the $x$-$y$ subspace. Diverse combinations are considered to provide an assessment in a wide operation range. The experimental tests for the evaluation of the time and frequency response of the scheme are described as follows.

First, the single frequency test with the maximum modulation index [10], $m_1 = 1.1547$ and $m_2 = 0$, is depicted in Figure 4. The traditional switching pattern and sinusoidal current can be appreciated in the time response. The voltage spectrum attained with the discrete Fourier transform from 0 to 5 (kHz) and 500 (Hz) per division is also included in capture of the oscilloscope. This shows a clean spectrum in the low-order frequencies interval. The harmonics components are only the effect of the modulation process in the sidebands around the carrier frequency ($m_f$-order harmonic component).

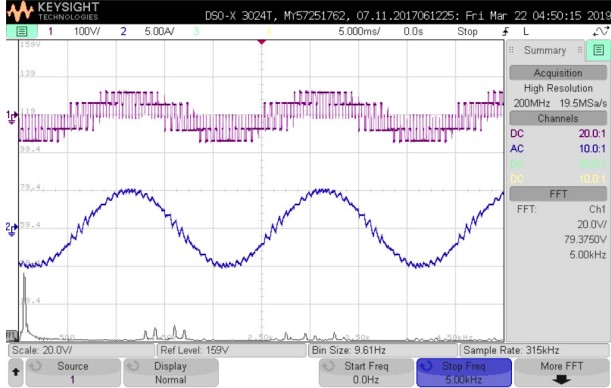

**Figure 4.** Voltage (top) and current (middle) waveforms of the phase $a$ along with the $V_a$ spectrum (bottom) for the reference voltage $m_1 = 1.15$ and $m_2 = 0$.

The proposed algorithm is capable of operating in the multifrequency mode. Then, the reference voltage of the modulator is set with a fundamental frequency of $m_1 = 0.92$ and a fifth-order harmonic of $m_2 = 0.23$. This combination provides the maximum dc-bus utilisation in the linear modulation region since $m_1 + m_2 = 1.1547$ [23]. The behaviour achieved is presented in Figure 5. The phase voltage has a slightly different switching pattern with respect to the previous case to synthesise the target voltage of the $x$-$y$ plane. The current waveform denotes the frequency mixing with non sinusoidal behaviour. On the other hand, the fast Fourier transform (FFT) proofs the low harmonic energy retained in the synthesised voltage. Furthermore, the sidebands around $2f_s$ are the more notable components, whereas for the single frequency this happens in the neighbourhood of the carrier frequency.

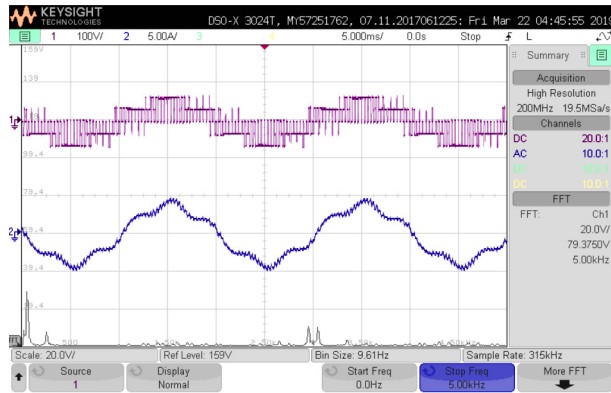

**Figure 5.** Voltage (top) and current (middle) waveforms of the phase *a* along with the voltage spectrum (bottom) for the reference voltage $m_1 = 0.92$ and $m_2 = 0.23$.

The most extreme combination for multiphase applications is $m_1 = m_2$. The response achieved for this case when both magnitudes are 0.57 to reach the maximum dc-bus utilisation is indicated in Figure 6. The voltage switching pattern is completely different respect to the conventional case, attaining a more dynamic waveform. The current behaviour in the time-domain also evidences the multifrequency operation. The frequency response again reproduces the previous features, generating undesired harmonics components only in the sidebands. The harmonic energy is practically distributed in equal proportions between $f_s$ and $2f_s$ in comparison with the last spectrum.

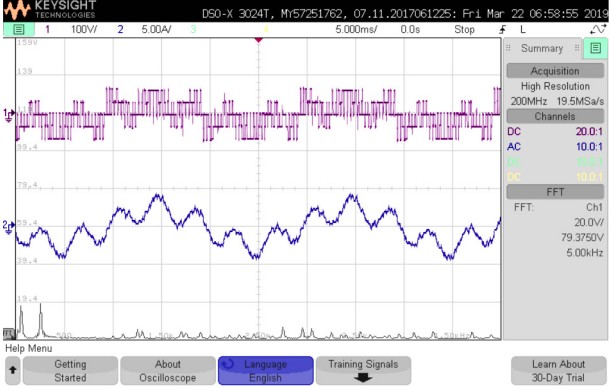

**Figure 6.** Time response of the voltage (upper graph) and current (middle waveform) of the phase *a* along with the voltage spectrum (bottom) for the reference voltage $m_1 = m_2 = 0.57$.

The control strategy in the synchronous frame to mitigate the dead-time effects of the real multiphase drive regulates the current of the *x-y* plane [18]. Therefore, the voltage that should be synthesised by the modulator in this use might be composed of a combination of the harmonic components that engage this subspace. An experimental test with simultaneous injection of the 5th- and 7th-order harmonics is carried out as a proof of concept of the proposal for the considered application, see Figure 7. Thus, the fundamental frequency is regulated with $m_1 = 0.90$ and the reference voltage in the *x-y* plane is the result of the mix between the two harmonics configured with magnitudes of 0.15 and 0.10, respectively. The response of the developed algorithm is appropriate for this case showing the expected waveforms, while the generation of fundamental and the two target harmonics components in agreement with their respective set-points are confirmed in the spectrum. Notice that the impact of the modulation process is low, and only the amplitude of the sidebands locate around 3 (kHz), $2f_s$, are perceptible.

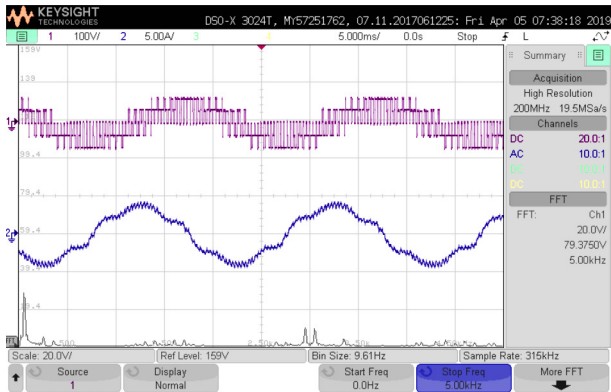

**Figure 7.** Voltage (upper graph) and current (middle waveform) of phase *a* along with the synthesised voltage spectrum (bottom) with a fundamental frequency magnitude of 0.90, while the 5th- and 7th-order harmonics are set to 0.15 and 0.10, respectively.

The magnitudes of the synthesised voltages in the previous tests are in agreement with the command values according to the obtained spectrum. Moreover, the time-domain response has the expected behaviour. Then, an analysis of the voltage and current distortion is the next step in this evaluation process. As two frequency components are injected, a compound total harmonic distortion (CTHD) is proposed to assess the method. This performance parameter is calculated as follows:

$$\text{CTHD}_f = \frac{\sqrt{\sum_{h \neq 1,5} f_h^2}}{\sqrt{f_1^2 + f_5^2}} \tag{10}$$

where *f* can be either the current or the voltage and the subscript *h* designates the *h*-order harmonic component. Then, this metric provides the proportion of the total amount of undesired harmonics achieved with respect to the injected magnitudes.

The ratio $r = m_2/m_1$ is defined to conduct the experimental results in this part. Then, *r* is adjusted with the following values 0.00, 0.10, 0.20, 0.40 and 1.00 to assess a wide range of operation points with the developed method. For each of these ratios, the exploration is performed by fixing *r* and varying the magnitude of $m_1$ from 0.30 up to the limit of the dc-bus utilisation, including also this last. The test rig is configured with these combinations of modulation indexes and the data waveform attained with the oscilloscope is stored to be processed. In this manner, the $\text{CTHD}_v$ and $\text{CTHD}_i$ are computed by applying (10) with the measurements.

The behaviour of the $\text{CTHD}_v$ achieved with the technique is illustrated in Figure 8. The single frequency tests provide the conventional total harmonic distortion [24], with decreasing values in all the intervals. The response of the scheme with ratios higher than zero shows similar characteristics. For $r < 0.50$ the $\text{CTHD}_v$ is practically identical to the first case, whereas for higher ratios the result is noticeably lower, see $r = 1.00$. The reason is the resulted sidebands of the spectrum. Notice that in the previous results of Figure 6, $m_1 = m_2$ and $r = 1$, the harmonic's energy is distributed around $f_s$ and $2f_s$, while for the other combinations, Figures 4 and 5, this is concentrated only in one of these frequencies. This effect generates a larger voltage harmonics that produce higher $\text{CTHD}_v$.

The measured currents are processed with (10) and the results are summarised in Figure 9. In all the cases, $\text{CTHD}_i$ decreases with respect to $m_1$. The notorious characteristic is the higher current distortion achieved for $r = 1.00$ in spite of reporting the lower $\text{CTHD}_v$ values in the previous test. Again, the frequency response of the voltage ought to be analysed to find the reasons. The voltage spectrum generated with $m_1 = m_2$ has smaller harmonic components, but they are distributed around $f_s$ and $2f_s$. In the other cases ($r < 0.50$) the concentration is in the neighbourhood of $2f_s$. Since the impedance is proportional to the frequency, the sidebands of $r = 1.00$ present lower opposition and generate higher harmonic currents, which impact the reckoning of $\text{CTHD}_i$.

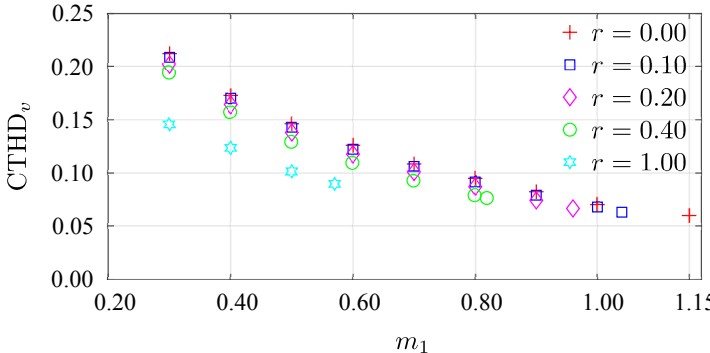

**Figure 8.** compound total harmonic distortion (CTHD$_v$) for different operation points within the linear modulation region.

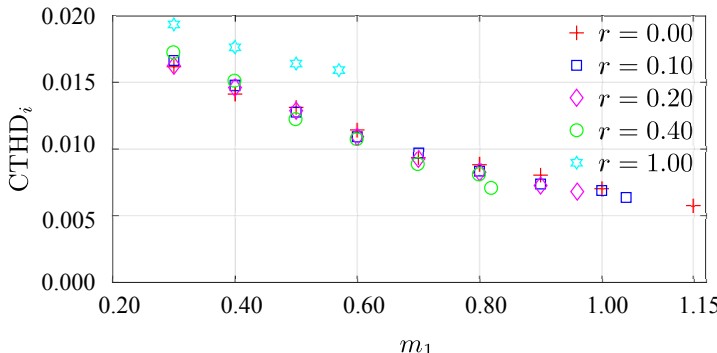

**Figure 9.** CTHD$_i$ for different operation points within the linear modulation region.

The time and spectrum behaviour of the multifrequency modulation strategy with continuous switching mode (two commutations per sampling period) was evaluated with more than 40 experimental tests. The degrees of freedom were configured to operate as the SVM for the sake of simplicity in the introduction of the technique. The undesired frequencies are present in the sidebands, caused by the mix between the carrier frequency, the fundamental and the harmonics injected in the *x*-*y* plane (modulation process). In this manner, the behaviour of the spectrum is available to design power filters or to mitigate possible sources of resonance, achieving robust electric drives. This section demonstrates the viability and great potential of the algorithm by following a rigorous procedure.

## 5. Conclusions

A multifrequency modulation technique for dual three-phase voltage source converters has been developed in this paper. The scheme is based on a generalised solution of the voltage-time law and implemented with a simple algorithm that computes the duty cycles of the power switches immediately. This output can be integrated straightforwardly with the pulse width modulation peripherals of the digital controllers. Validation tests to analyse the time and frequency response were conducted by injecting fundamental and fifth-order frequency components. The proposal demonstrates good magnitude tracking with a small number of undesired harmonic components, which are generated due to the modulation process. The current distortion attained (associated with the current ripple) is also low and was measured with a proposed figure of merit. The promising features of the method for multiphase power electronics applications can be proved in this manner.

**Author Contributions:** Conceptualization, J.A.R. and J.P.; methodology, J.A.R., M.R. and R.G.; software, J.A.R., S.T. and M.R.; validation, J.P. and J.A.R.; formal analysis, R.G., M.R. and J.P.; investigation, J.A.R., J.P. and M.R.; resources, M.R., R.G. and S.T.; data curation, M.R. and R.G.; writing—original draft preparation, J.A.R., M.R. and S.T.; writing—review and editing, R.G., M.R., S.T. and J.P.; visualization, J.A.R., M.R. and S.T.; supervision, J.A.R. and R.G.; project administration, R.G., J.A.R. and M.R.; funding acquisition, S.T., R.G., M.R. and J.A.R.

**Funding:** This research was funded in part by Paraguayan Program for the Development of Science and Technology (PROCIENCIA) through the R&D project component with reference 14-INV-097, in part by the Chilean Fund of Scientific and Technological Development (FONDECYT) under the grants provided in the Postdoctoral 3170014 and the Regular 1160690 projects.

**Conflicts of Interest:** The authors declare no conflict of interest.

## Appendix A. Generalised Modulation Scheme for Three-Phase VSC

A three-phase modulation space described by the switching signals, and represented by using the constants previously defined in this paper, is given by [22]:

$$
\begin{aligned}
v_d &= (4/3)(S_a - S_\delta S_b - S_\delta S_c) \\
v_q &= (4/3)(C_\delta S_b - C_\delta S_c)
\end{aligned}
\tag{A1}
$$

where $v_d$ and $v_q$ are the normalised voltages space vector components in the stationary reference frame generated by the switching vector $[S_a\ S_b\ S_c]$. Then, the modulation law of this system described with the normalised duty cycles $t_j = T_j/T_s$ of the switching signals along with the reference voltages $v_d^*$ and $v_q^*$ is presented as follows:

$$
\begin{aligned}
v_d^* &= (4/3)(t_a - S_\delta t_b - S_\delta t_c) \\
v_q^* &= (4/3)(C_\delta t_b - C_\delta t_c)
\end{aligned}
\tag{A2}
$$

The system of linear equation attained in (A2) could have an infinite number of solutions within the domain of $t_j \in [0, 1]$ and the linear modulation region. The geometric interpretation is the intersection line of two three-dimensional planes, which can be determined by an appropriate transformation in a two-dimensional auxiliary space. A generalised analytical solution was proposed in [22] by adding the $\lambda \in [0, 1]$ degree of freedom, that is the zero-sequence voltage of the electric modelling. Thus, the duty times can be computed by applying Algorithm A1.

---

**Algorithm A1:** Generalised PWM for three-phase inverters

---

    **Inputs:** $v_d^*, v_q^*, \lambda$

    **Constants and variables:** $C_\delta, S_\delta, \tau_d, u^*, \tau_{11}, a$

    **Outputs:** $t_a, t_b, t_c$

    // Calculate the algorithm constants

    $\tau_d = C_\delta \cdot |v_q^*|$

    $u^* = C_\delta^2 \cdot v_d^* + S_\delta \cdot \tau_d$

    // Calculate the values of $\tau_{11}$ and $a$

    **if** $(-C_\delta \leq u^* \leq 0)$ **then**

        $\tau_{11} = 0$

        $a = u^* + 1 - \tau_d$

    **else if** $(0 < u^* \leq \tau_d)$ **then**

        $\tau_{11} = u^*$

        $a = 1 - \tau_d$

    **else if** $(\tau_d < u^* \leq 1)$ **then**

        $\tau_{11} = u^*$

        $a = 1 - u^*$

    **end if**

    //Calculate the duty cycles

    $t_a = \tau_{11} + a \cdot \lambda$

    $t_b = t_a - C_\delta \cdot (C_\delta \cdot v_d^* - S_\delta \cdot v_q^*)$

    $t_c = t_b - C_\delta \cdot v_q^*$

---

**Table A1.** Operation modes of the generalised pulse width modulation (PWM)

| $\lambda$ | PWM Technique |
|---|---|
| 0 | PWM-Min |
| 1/2 | SVM |
| 1 | PWM-Max |

The inputs of the presented algorithm are the normalised reference voltages and the zero-sequence control variable. Then, the auxiliary constants $\tau_d$ and $u^*$ are calculated with simple operators. These are processed by a conditional routine with three possible cases to attain the auxiliary values of $\tau_{11}$ and $a$, respectively. These are used to compute $t_a$, whereas this value is applied into (A2) to solve the linear equation system and obtain the remaining duty cycles. Notice that the architecture is simple and only employs low computational cost operations.

The degree of freedom $\lambda$ has been identified as the proportion between the dwell time of the vector [1 1 1] and the total application time of the zero voltage vector, which is also distributed with the vector [0 0 0]. Hence, the generalised modulator can be configured to operate like the well-known techniques [25] by selecting the values of $\lambda$ presented in Table A1.

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
