# Peer review of "A Generalised Multifrequency PWM Strategy for Dual Three-Phase Voltage Source Converters"

_energies, doi:10.3390/en12071398_

Round 1

Reviewer 1 Report

The paper presents a generalised Multifrequency PWM strategy for dual three-phase voltage source converters.

Introduction section is complete. It includes four key components: motivation, literature survey, contributions, and the organization of paper.

Section 2, 3, 4,  and 5 are complete. The paper is well structured.

In conclusion the paper has its summary, the modulation technique for dual three phase voltage source converter was validated with tests to analyse the time and frequency response by injecting a higher order frequency components in the modulation.

As it is clearly presented the promising features of the converter are outlined. A short comment may be added regarding the implementation on DSP and interaction with FPGA.

Author Response

Dear Reviewer,

Thank you for your comments. Please, find attached a pdf file with a detailed response to your suggestions.

Best regards,

The authors.

Reviewer 2 Report

Authors should discuss the risk of resonance in the DC-link capacitor(s)

One of the reasons to adopt multi-three-phase solution is the fault tolerance: how this solution passes form the complete modulation to the 3-phase only if one of the start converter fails?

Author Response

(The authors gave the same response as above.)

Reviewer 3 Report

It is an interesting paper to use six half-bridge legs to achieve an inverter function. It is very helpful to achieve the redundancy and increase the reliability of the whole system. It is a very good work with experimental validation. The reviewer would like to approve it. The only small suggestions are:

1.       Have a detailed comparison with the 4-legs inverter. The cost of this system is absolutely higher. The author should do a comparison with the existing systems.

2.       Fig. 3 shows the prototype. However, it is not clear. Please show more details, including the circuit board hardware and the connection between boards.

3.       Only three waveform figures are provided, which is not sufficient. Please provide more high power experimental wavefroms.

Author Response

Dear Reviewer,

Thank you for your comments. Please, find attached a pdf file with a detailed response to your suggestions.

Best regards,

The authors
